# Associations of Total Body Fat Mass and Skeletal Muscle Index with All-Cause and Cancer-Specific Mortality in Cancer Survivors

**DOI:** 10.3390/cancers15041081

**Published:** 2023-02-08

**Authors:** Livingstone Aduse-Poku, Shama D. Karanth, Meghann Wheeler, Danting Yang, Caretia Washington, Young-Rock Hong, Todd M. Manini, Jesus C. Fabregas, Ting-Yuan David Cheng, Dejana Braithwaite

**Affiliations:** 1Department of Epidemiology, University of Florida, 2004 Mowry Rd., Gainesville, FL 32610, USA; 2University of Florida Health Cancer Center, University of Florida, 2004 Mowry Rd., Gainesville, FL 32610, USA; 3Aging & Geriatric Research, Institute on Aging, 2004 Mowry Rd., Gainesville, FL 32610, USA; 4Department of Health Services Research, Management, & Policy, 1225 Center Dr., Gainesville, FL 32610, USA; 5Department Health Outcomes & Biomedical Informatics, Institute on Aging, 2004 Mowry Rd., Gainesville, FL 32611, USA; 6Division of Hematology & Oncology, College of Medicine, University of Florida, 2000 SW Archer Rd., Gainesville, FL 32608, USA; 7Division of Cancer Prevention and Control, Department of Internal Medicine, The Ohio State University, Columbus, OH 43210, USA; 8Department of Surgery, University of Florida, 1600 SW Archer Rd., Gainesville, FL 32608, USA

**Keywords:** total fat mass, sarcopenia, all-cause mortality, skeletal muscle index, and race

## Abstract

**Simple Summary:**

Studies assessing the associations of body composition and mortality have found conflicting results due to small sample sizes and short follow-up periods. Using a nationally representative sample, we assessed the associations between total body fat mass and skeletal muscle index, and all-cause and cancer-specific mortality in cancer survivors. Participants with higher fat mass and sarcopenia were at a higher risk of mortality. Our results have significant clinical and public health implications, since fat and muscle mass are important prognostic factors in cancer patients. The findings also highlight the importance of healthy body composition and the need for more research into the effects of specific interventions, such as physical activities, a healthy diet, and supplemental nutrition aimed at reducing fat mass and preventing muscle loss.

**Abstract:**

**Purpose**: The importance of body composition on cancer outcomes is of great clinical interest. Measures of body composition that differentiate fat mass from skeletal muscle mass can help redefine our understanding of body composition for cancer survival. We investigated whether the risk of all-cause and cancer-specific mortality differ by levels of total fat mass and sarcopenia status in cancer survivors. Our secondary aim was a subgroup analysis assessing the role of race within these associations. **Methods**: Participants included 1682 adult cancer survivors who had undergone a dual-energy X-ray absorptiometry (DXA) examination to measure body composition, from the 1999–2006 and 2011–2018 National Health and Nutrition Examination Survey (NHANES). Total fat mass was categorized into tertiles (we assessed high vs. low tertiles), and sarcopenia was considered as having an appendicular skeletal muscle mass index less than 7.26 kg/m^2^ for males and less than 5.45 kg/m^2^ for females. Multivariable Cox proportional hazard models estimated the adjusted hazard ratio (aHR) and 95% confidence interval (CI). **Results:** The mean age of study participants was 61.9 years, and they were followed up for an average of 9.67 years. The prevalence of sarcopenia was 25.0% (N = 304), and 33.4% (N = 561) had a high total fat mass. Participants with a higher fat mass (aHR = 1.30, 95% CI = 1.06–1.61) and with sarcopenia (aHR = 1.51, 95% CI = 1.22–1.88) had a 30% and 51% increased risk of all-cause mortality compared to participants with a low fat mass and with no sarcopenia, respectively. Further, sarcopenia (aHR = 1.74, 95% CI = 1.23–2.29) was associated with a higher risk of cancer-specific mortality in cancer survivors. The association between sarcopenia and all-cause mortality was twice as strong in Black people (aHR = 2.99, 95% CI = 1.39–6.06) compared to White people (aHR = 1.53, 95% CI = 1.19–1.95). **Conclusions:** Our findings show the opposing relations of fat mass and appendicular skeletal muscle mass index with mortality in a national sample of cancer survivors, and that the relationships may differ by race. These results emphasize the importance of maintaining a healthy body composition among cancer survivors.

## 1. Introduction

Obesity and being overweight (BMI ≥ 25 kg/m^2^) are common risk factors for most cancers. According to the National Cancer Institute (NCI) and the International Agency for Research on Cancer (IARC), obesity and being overweight increase the risk of at least 13 cancers, including breast, colorectal, pancreatic, stomach, gallbladder, liver, thyroid, ovarian, endometrial, kidney, multiple myeloma, esophageal, and meningeal cancers [1,2]. However, patients with a higher body mass index (BMI) among the overweight to moderate obesity range (>25 kg/m^2^–<35 kg/m^2^) paradoxically have a lower overall mortality rate after a cancer diagnosis, a phenomenon known as the “obesity paradox” [3]. Most epidemiological studies use BMI to measure overweight and obesity. However, BMI does not distinguish between fat mass and lean mass [4]. In addition, a low BMI may mask excess fat, and a high BMI may mask high lean mass [4]. To overcome these limitations, recent studies recommend using imaging techniques, such as dual-energy X-ray absorptiometry (DXA) and computed tomography (CT), to measure body composition [3,5].

DXA takes advantage of the difference in the X-ray attenuation of tissue and bone at different X-ray energies in order to measure lean body mass (LBM), fat mass (FM), and bone mineral mass (BMM), which can be extrapolated to the entire body. DXA has been used in many epidemiological studies for body composition measurements since it produces less radiation exposure and is less expensive than other methods, such as CT and magnetic resonance imaging (MRI) [5,6,7]. Body composition is an important measure after a cancer diagnosis. It measures components such as fat mass and skeletal muscle mass index. Skeletal muscle index is associated with the poor prognosis of various cancers and chemotherapy-related toxicities [8,9,10,11]. Both fat mass and skeletal muscle index have been observed to be associated with survival in cancer patients. Although most of the recent literature examining the association between body composition, all-cause mortality (death due to any cause) and cancer-specific mortality (deaths with cancer as the underlying cause), and using imaging techniques, has found conflicting results, most studies observed that high skeletal muscle mass and low adiposity levels were associated with increased survival in cancer patients [4,12,13,14,15]. Potential explanations for the variation are that most studies in this area had small sample sizes, short follow-ups, and a select population.

One critical but underexplored area of obesity research is on the impact of race on the association between body composition and mortality. Previous studies have observed racial differences in fat mass and skeletal muscle mass [16,17]. There are very few studies evaluating the association between body composition and mortality by race, and the majority of those studies have used BMI to assess body composition [18,19]. These studies observed no significant difference in the association between body composition and mortality among Black people and White people [18,19]. In our study, we aim to examine the associations between DXA-derived total fat mass and appendicular skeletal muscle mass index, and overall and cancer-specific mortality in cancer survivors. We will also investigate whether the association between these measures of body composition and mortality differ by race.

## 2. Materials and Methods

The National Health and Nutrition Examination Survey (NHANES) is a multistage, cross-sectional survey conducted in the United States (U.S.); NHANES is designed to monitor the nutritional status and health of a civilian, non-institutionalized U.S. population. Participants from approximately 30 U.S. counties are selected every couple of years to complete a personal, structured interview at home, and undergo standardized laboratory examinations and laboratory tests in mobile health centers. NHANES questionnaires cover a multitude of topics, including demographic characteristics, general health, lifestyle, and disease history. In these secondary analyses, we used 1999–2006 and 2011–2018 NHANES data, which were linked to death certificate records from the National Death Index (NDI). Follow-up began at the time of the baseline interview and continued until death, last contact, or 31 December 2019. The eligibility criteria for this study included participants who (1) underwent DXA examination to measure body composition, (2) had cancer history, and (3) had survived for at least 1 year after cancer diagnosis, since the acute effects of cancer treatment substantially impact nutrition-related measures [20]. A total of 1682 cancer survivors were included in the present study. Additional details regarding participant selection are in the Figure 1.

### 2.1. Exposures of Interest

Whole-body dual-energy X-ray absorptiometry (DEXA) scans were used to measure total fat mass and appendicular skeletal mass, which consists of the upper and lower limbs’ muscle mass. Total fat mass was categorized into tertiles (low, medium, and high) with 7094.3–23,548.8 g, 23,548.9–32,209.9 g, and 322,010–99,662.0 g as cut-offs for low, moderate, and high fat mass, respectively. We assessed high vs. low total fat mass. A standard protocol [21] was used by trained and certified radiology technologists to perform the DEXA scans. The appendicular skeletal muscle mass was used to determine the skeletal muscle index (ASMI), calculated by skeletal muscle mass/height squared (ASM/height^2^) in kg/m^2^. Appendicular skeletal muscle mass was calculated using the sum of the lower and upper limbs [22]. A trained stadiometer measured the participants’ height without shoes. The European Working Group on Sarcopenia in Older People (EWGSOP) criteria (i.e., ASMI less than 5.45 kg/m^2^ for females and less than 7.26 kg/m^2^ for males) were used to define sarcopenia [20]. We also included other proxy measures for body composition, such as body mass index (BMI) and waist circumference (WC).

### 2.2. Outcomes

The outcomes of interest were (1) cancer all-cause mortality, (2) cancer-specific mortality, and (3) obesity-related cancer mortality. Obesity-related cancer mortality was defined as adjudicated deaths attributed to breast, colorectal, pancreatic, stomach, gallbladder, liver, thyroid, ovarian, endometrial, kidney, multiple myeloma, or esophageal cancer, according to the National Cancer Institute [2]. Death was identified by linkage to the NDI through 31 December 2019. The cause of death was ascertained by the International Classification of Diseases, Tenth Revision (ICD-10).

### 2.3. Other Covariates

Demographic characteristics, such as age (<50, 50–64, 65–74, or ≥75 years), race (non-Hispanic white people, non-Hispanic Black people, or others, including Asians, Mexican Americans and multiracial groups), sex (male or female), education (high school or less, attended college, or graduated from college), and marital status (not married or married) were obtained through self-report. Participants were categorized as never smokers, current smokers, or former smokers. Smoking has been seen to increase the risk of sarcopenia [5]. Daily energy intake was measured in 24-h food recall at the baseline interview and was categorized into approximate quartiles (<1360 kcal/day, 1361 to 1785 kcal/day, 1786 to 2348 kcal/day, >2349 kcal/day). In our study, we incorporated the histories of 4 self-reported diseases (heart attack, stroke, coronary heart disease, and congestive heart failure). The time that had elapsed since cancer diagnosis was categorized into (1–3, 4–6, ≥7) years. Study variables were selected based on prior knowledge of the relationship between the exposures and outcomes of interest [6,7,8].

### 2.4. Statistical Analysis

We summarized the distribution of the cancer survivors’ demographic characteristics by tertiles of total fat mass. We then elucidated the number of deaths, person-year during follow-up, and mortality rate among the cancer survivors. Age and multivariable-adjusted hazard ratios (aHRs), as well as the corresponding 95% confidence intervals (CIs) for the risk of all-cause and cancer-related mortality associated with tertiles of fat mass and presence of sarcopenia, were computed using Cox proportional hazard models. When the model’s outcome was all-cause mortality, participants who were classified as either alive or deceased from non-cancer-related causes were censored. Multiple covariates were adjusted for in the proportional hazard models, such as age, sex, education, marital status, race, smoking, energy intake, and comorbidity burden. In addition, we further corrected for NHANES sampling weight. There was no violation of the proportional hazards assumption when assessed based on scaled Schoenfield residuals.

Subgroup analyses were based on age (<50 vs. ≥50 years), sex (female vs. male), energy intake (<1785 vs. ≥1785 kcal/day), and comorbidities (0 vs. ≥1 comorbidity). A Wald test was used to identify whether the interaction between these factors and the outcomes of interest were significant. Kaplan–Meier curves were used to visualize risk for all-cause and cancer-specific mortality by total fat mass and sarcopenia in cancer survivors. In addition, we stratified by (i) cancer types according to whether or not they were related to obesity and (ii) race (Non-Hispanic White people and Non-Hispanic Black people) and examined the associations of total fat mass and sarcopenia with all-cause and cancer-specific mortality. To assess the potential nonlinear associations of total fat mass and sarcopenia with all-cause mortality, and to capture the variation in risk across the entire continuum of the relations, we used restricted cubic splines with 4 knots. Two-sided p-values of less than 0.05 determined the statistical significance. These statistical analyses were performed using SAS v9.4 (SAS Institute Inc., Cary, NC, USA) and R statistical software (version 4.0.2).

## 3. Results

In our study, the median follow-up time was 9.7 years. Participants included 1682 adult cancer survivors (Figure 1). A total of 668 (39.7%) of the cancer survivors died during follow-up. Cancer-specific deaths were observed in the 213 (15.6%) of the study participants. The mean age of the study participants was 61.9 years. In our study sample, the prevalence of sarcopenia was 25.0% (N = 304) (Table 1). The mean total fat mass was 29,400 (range = 7094–99,662) and mean ASMI was 7.28 (range = 3.74–13.91 kg/m^2^). Most of the participants were males, 56.2% (N = 945), and White 72.1% (N = 1213). Participants with high school or less education had a higher prevalence of total fat mass compared to participants who had graduated from college (48.8% vs. 25.0%). Participants who were current smokers had a higher total fat mass compared to former smokers (35.2% vs. 4.1%).

In the age-adjusted Cox proportional hazards regression models (Table 2), high total fat mass (Adjusted Hazard Ratio; aHR = 1.76, [95% CI = 1.45–2.13], model 1) and sarcopenia (aHR = 1.58, [95% CI = 1.28–1.94], model 1) were associated with a higher risk of all-cause mortality. In the multivariable Cox models (Table 2), a higher total fat mass (aHR = 1.30, [95% CI = 1.06–1.61], model 2) and sarcopenia (aHR = 1.51, [95% CI = 1.22–1.88], model 2) were associated with a higher risk of all-cause mortality in cancer survivors. Correction for NHANES sampling weight did not substantially alter the results. In addition, in the age-adjusted Cox proportional hazards regression model, a high total fat mass (aHR = 1.42, [95% CI = 1.01–2.00], model 1) and sarcopenia (aHR = 1.88, [95% CI = 1.33–2.65], model 1) were associated with a higher risk of cancer-specific mortality. In the multivariable model Cox models, sarcopenia (aHR = 1.74, [95% CI = 1.23–2.29], model 2) was associated with a higher risk of cancer-specific mortality in cancer survivors; however, total fat mass was not associated with cancer-specific mortality.

For Table 3, participants were classified according to whether or not they had obesity-related cancers. In the age-adjusted Cox proportional hazard models, total fat mass (aHR = 1.81, [95% CI = 1.34–2.43], model 1) and sarcopenia (aHR = 1.49, [95% CI = 1.09–2.03], model 1) were associated with a higher risk of all-cause mortality in obesity-related cancer survivors; however, total fat mass was not associated with all-cause mortality in non-obesity-related cancer survivors. The results did not change significantly after adjusting for the covariates. There was no association between total fat mass and the risk of mortality in obesity-related and non-obesity-related cancer patients. In the multivariable Cox models, sarcopenia (aHR = 1.76, [95% CI = 1.03–3.02], model 2) was associated with a higher risk of all-cause mortality in obesity-related cancer survivors. When participants were stratified according to race, high total fat mass was associated with an increased risk of mortality after adjusting for covariates (aHR = 1.37, [95% CI = 1.07–1.75], model 2, Table 4); however, total fat was not associated with all-cause mortality in Black people. The association between sarcopenia and all-cause mortality was twice as strong in Black people (aHR = 2.99, [95% CI = 1.39–6.06], model 2) compared to Whites (aHR = 1.53, [95% CI = 1.19–1.95], model 2).

Kaplan–Meier curves suggested that participants with a high total fat mass (Figure 2A) and those with sarcopenia (Figure 2B) had a higher risk for all-cause mortality. In subgroup analyses for cancer survivors (Appendix A), we found that total fat mass significantly interacts with sex and age in relation to all-cause mortality. Specifically, the effect size of total fat mass was significantly larger in males (female: aHR = 1.53, 95% CI = 0.92–2.55; male: aHR = 2.16, 95% CI = 1.15–4.06; *p*-interaction < 0.001). For Appendix A, a higher total fat mass (aHR = 1.33, [95% CI = 1.03–1.72], model 2) and sarcopenia (aHR = 1.46, [95% CI = 1.11–1.91], model 2) were associated with a higher risk of non-cancer mortality after adjusting for covariates. Also, we observed a J-shaped positive association between total fat mass and all-cause mortality (Figure 3).

## 4. Discussion

In this study, we used a nationally representative sample of 1682 cancer survivors with a mean age of 61.9 years, who were followed up for an average period of 9.67 years. We identified positive associations between both total fat mass and sarcopenia, and all-cause and cancer-specific mortality. Participants with a higher fat mass and sarcopenia had a 30% and 51% higher risk of all-cause mortality risk, respectively. Sarcopenia was also associated with a higher risk of cancer-specific mortality; however, total fat mass was not associated with cancer-specific mortality. In addition, we found that a high total fat mass was associated with an increased risk of mortality in White people, but not in Black people. The association between sarcopenia and all-cause mortality was about twice as strong in Black people compared to White people. Among obesity-related cancer survivors, total fat mass was associated with a higher risk of all-cause but not cancer-specific mortality; however, among non-obesity cancer survivors, total fat mass was not associated with either all-cause or cancer-specific mortality. The outcomes of the subgroup analyses indicate that the magnitude of the association between total fat mass and risk of all-cause death was stronger in male than in female cancer survivors.

While previous studies examining the association between fat mass and all-cause mortality have found inconsistent results, most have observed a J- or U-shaped association between fat mass and mortality [23,24,25,26,27]. Other studies have identified an inverse relationship between fat mass and all-cause mortality [28,29], while some studies found no association [30,31,32]. In this study, we observed a J-shaped positive association between total fat mass and all-cause mortality. Although studies examining the association between total fat mass and mortality among cancer survivors are limited, several studies have observed direct positive associations between subcutaneous, visceral adipose tissues (VAT) [33,34] and total adipose tissue [4], and all-cause mortality, which is consistent with our findings. Nevertheless, meta-analysis of six studies examining the association between VAT and overall survival among prostate cancer patients found no association [35]. In addition, in a recently published meta-analysis of 128 studies, En Cheng and colleagues found that images measuring visceral, subcutaneous, and total adiposity were not associated with an increased risk of mortality among cancer survivors; however, high heterogeneity was observed among cancer types [36]. Reasons for the discrepancies in the fat mass-mortality association in the literature include the restriction to an elderly population [28,29,30], short follow-up periods [14,15,23,37], failure to adjust for important confounders [15,32,38], failure to assess the shape of the association using spline or other techniques [29,32,39], lack of mutual adjustment for fat mass [15,40], use of different techniques to assess fat mass, and measurement errors.

An important aspect of the fat mass-mortality association is whether the relationship differs by race. Previous studies assessing racial differences in the fat mass-mortality relationship have used BMI as a measure of body composition, despite its limitations. In this study, we found that high total fat mass was associated with an increased risk of mortality in White people; however, total fat mass was not associated with all-cause mortality in Black people. Numerous studies have assessed the association between BMI and mortality in White people [41,42,43,44,45,46,47,48]. In contrast, few studies have evaluated the BMI–mortality association in Black people [49,50,51]. Of those which have, most studies have observed a stronger BMI–mortality association in White people compared to Black people [41,51]. A few studies compared the BMI–mortality association in both Black people and White people. One of those studies was a large prospective study by Calle and colleagues; in this study, the association between BMI and mortality appeared to be weaker in Black people compared to White people after a 14-year follow-up [52]. Another large prospective study observed that the magnitude of the BMI–mortality association was similar in White people and Black people. In other studies, although Black people have been known to have a lower prevalence of sarcopenia than White people, Black people have a higher mortality rate [53]; other studies have shown that socioeconomic factors (such as income and educational status) may play a significant role in this observation [54,55]. Several physiological mechanisms underlying the association of fat mass with tumor progression and mortality have been proposed. High fat mass may lead to high circulating insulin and/or insulin-like growth factor 1, altered levels of leptin, and tissue-level inflammation. These factors downgrade anti-tumor immunity and increase the risk of tumor angiogenesis, growth, and metastasis. Metastasis is known to be responsible for 90% of cancer deaths [56].

Skeletal muscle mass is an important prognostic factor among cancer patients as it is associated with the quality of muscles, activity levels, and treatment-related toxicities. In this study, sarcopenia was measured by the skeletal muscle mass index and was found to be associated with all-cause mortality and cancer-related mortality in cancer survivors. In line with this finding, a higher skeletal muscle mass has been observed to increase the risk of mortality among prostate cancer [57], colorectal cancer [58,59,60], breast cancer [4,61], and advanced cancer [13] patients. In addition, skeletal muscles may be associated with insulin growth factor signaling, host immunity, systematic inflammation, and hormonal activity, which are related to poor cancer prognosis [62]. In addition, sarcopenia may indicate a progressive withdrawal of anabolism and increased catabolism [63], which are linked to functional decline, frailty, and a shorter life expectancy [64].

Our study has several strengths in design and analysis. First, it was conducted in a nationally representative sample. Second, total fat mass and sarcopenia were measured by DEXA. Third, vital status and cause of death were assessed by linkage to NDI and ICD codes, respectively, ensuring the validity of exposure and outcome of interest. Fourth, we also used other proxy measures of fat mass, such as BMI and waist circumference, to assess their associations with mortality. Finally, we assessed the shape of the association and dose–response relationship between the predictors and outcomes using restricted cubic splines in the Cox proportional hazards regression models. However, the results from this study should be interpreted in light of the following limitations. Most covariates, such as age, cancer types, smoking status, and comorbidity, were all self-reported, making them less valid compared to medical records due to their higher likelihood of recall bias. We also did not use a control for cancer treatment. Since cancer treatment can affect fat and skeletal muscle mass, as well as mortality, not controlling for this variable and other potential confounders could have led to residual confounding. In addition, the study has a relatively small sample and short follow up period of less than 10 years. In addition, since this is an observational study, we cannot infer causation.

## 5. Conclusions

In this nationally representative sample of US adults, we found opposing relations between fat mass, appendicular skeletal muscle mass and mortality. Moreover, we found that the body composition–mortality association differed by race. Fat and skeletal muscle mass may act differently on health outcomes, including mortality, which has long been used to explain the existence of the “obesity paradox”. Our results have significant clinical and public health implications, since fat and muscle mass are important prognostic factors in cancer patients. The findings also highlight the importance of healthy body composition, instead of focusing on weight alone. There is a need for more research into the effects of specific interventions, such as physical activities, a healthy diet, and supplemental nutrition aimed at reducing fat mass and preventing muscle loss.

## Figures and Tables

**Figure 1 cancers-15-01081-f001:**
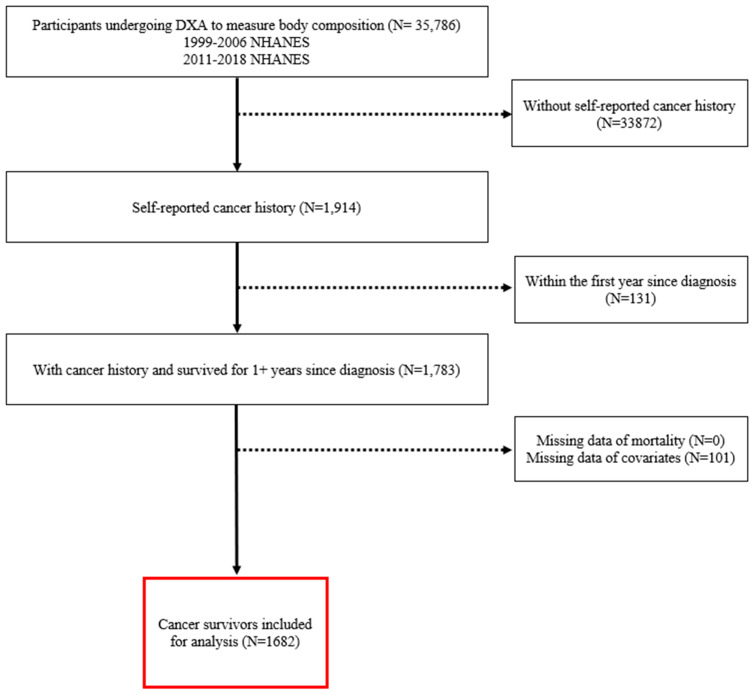
Participants selection diagram.

**Figure 2 cancers-15-01081-f002:**
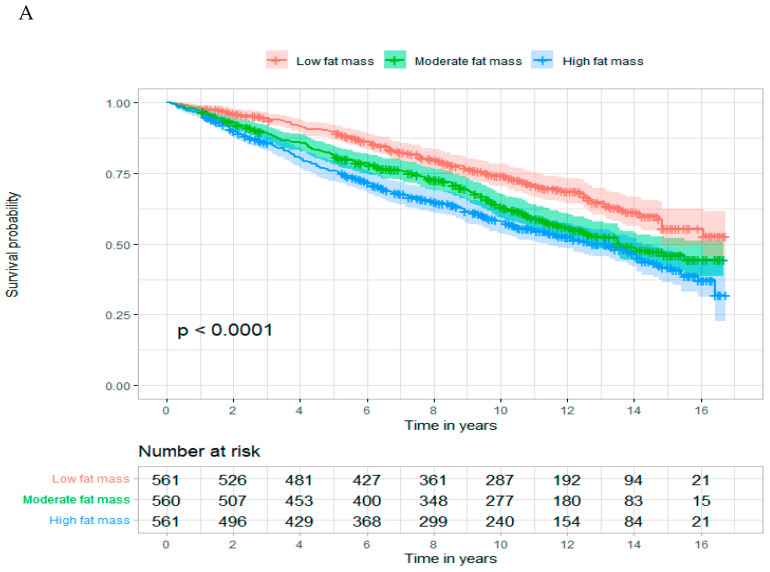
(**A**). Kaplan–Meier curve for total fat mass and all-cause mortality; (**B**). Kaplan–Meier curve for sarcopenia and all-cause mortality.

**Figure 3 cancers-15-01081-f003:**
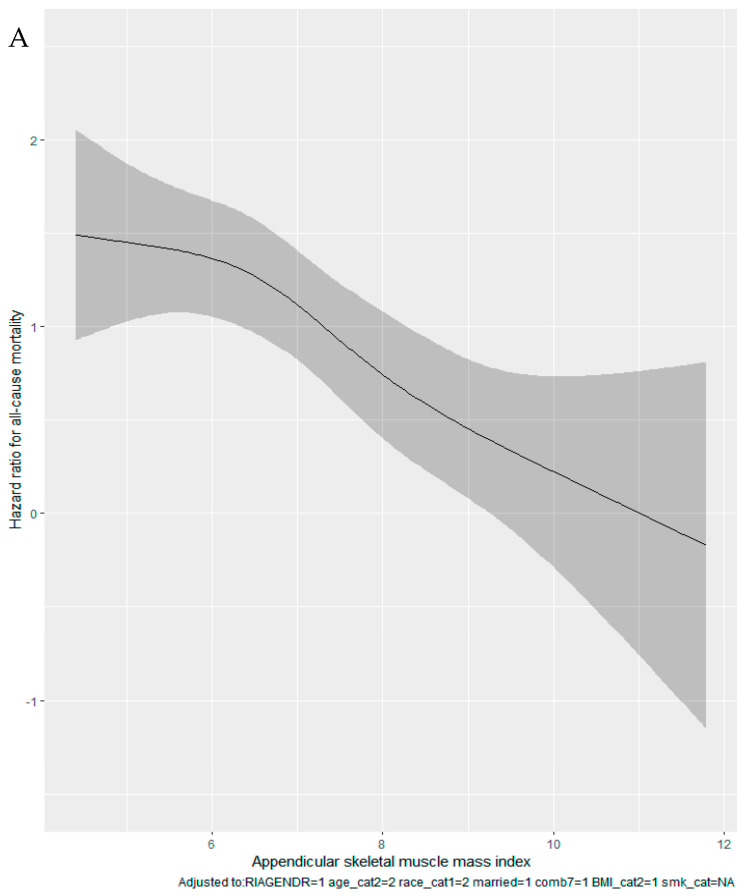
(**A**) Spline functions with corresponding 95% CIs from Cox proportional hazards regression for the relations of sarcopenia; (**B**) Spline functions with corresponding 95% CIs from Cox proportional hazards regression for the relations of total fat mass.

**Table 1 cancers-15-01081-t001:** Summary of study characteristics, N = 1682.

	Cancer Survivors (N = 1682) No. (%)			
	Overall	Low Total Fat Mass	Moderate Total Fat Mass	High Total Fat Mass
**Participant Characteristics**	**1682 (100)**	**561 (33.3)**	**560 (33.3)**	**561 (33.3)**
**Age (year)**				
<50	402 (23.9)	161 (28.7)	101 (18.0)	140 (25.0)
50–64	421 (25.0)	169 (30.1)	127 (22.7)	125 (22.3)
65–74	426 (25.3)	150 (26.7)	154 (27.5)	122 (21.7)
≥75	433 (25.7)	81 (14.4)	178 (31.8)	174 (31.0)
**Sex**				
Female	737 (43.8)	164 (29.2)	255 (45.5)	318 (56.7)
Male	945 (56.2)	397 (70.8)	305 (54.5)	243 (43.3)
**Race**				
White	1213 (72.1)	380 (67.7)	410 (73.2)	423 (75.4)
Black	216 (12.8)	94 (16.8)	58 (10.4)	64 (11.4)
Other	253 (15.0)	87 (15.5)	92 (16.4)	74 (13.2)
**Education**				
High school or less	810 (48.2)	272 (48.5)	265 (47.3)	273 (48.8)
Attended college	473 (28.2)	179 (31.9)	148 (26.4)	146 (26.1)
Graduated from college	397 (23.6)	110 (1.6)	147 (26.3)	140 (25.0)
**Marital status**				
Not married	672 (40.6)	235 (42.4)	215 (39.2)	222 (40.1)
Married or living with partner	983 (59.4)	319 (57.6)	333 (60.8)	331 (59.9)
**Smoking status**				
Never	664 (68.2)	223 (70.3)	229 (74.4)	212 (60.7)
Current	275 (28.2)	82 (25.9)	70 (22.7)	123 (35.2)
Former	35 (3.6)	12 (3.8)	9 (2.9)	14 (4.0)
**BMI**				
Underweight	28 (2.1)	28 (6.2)	0 (0.0)	0 (0.0)
Normal	411 (30.7)	307 (67.9)	103 (22.4)	1 (0.2)
Overweight	478 (35.8)	116 (25.7)	284 (61.7)	78 (18.4)
Obese	420 (31.4)	1 (0.2)	73 (15.9)	346 (81.4)
**Energy intake (kcal/day)**				
<1359.0	345 (26.4)	117 (28.0)	114 (25.5)	114 (25.9)
1360.0–1784	323 (24.7)	107 (25.6)	112 (25.1)	104 (23.6)
1785–2348	354 (27.1)	112 (26.8)	120 (26.8)	122 (27.7)
≥2349.0	284 (21.7)	82 (19.6)	101 (22.6)	101 (22.9)
**ASMI**				
Without sarcopenia	913 (75.0)	326 (88.8)	328 (78.3)	259 (60.1)
With sarcopenia	304 (25.0)	41 (11.2)	91 (21.7)	172 (39.9)
**No. comorbidities**				
0	519 (30.9)	135 (24.1)	178 (31.8)	206 (36.7)
1	450 (26.8)	164 (29.2)	156 (27.9)	130 (23.2)
≥2	713 (42.4)	262 (46.7)	226 (40.4)	225 (40.1)
**Stroke**				
No	1566 (93.3)	525 (93.6)	522 (93.7)	519 (92.5)
Yes	113 (6.7)	36 (6.4)	35 (6.3)	42 (7.5)
**Coronary heart disease**				
No	1215 (89.8)	395 (91.4)	418 (89.7)	402 (88.4)
Yes	138 (10.2)	37 (8.6)	48 (10.3)	53 (11.6)
**Congestive heart failure**				
No	1252 (91.5)	401 (90.7)	431 (91.7)	420 (91.9)
Yes	117 (8.5)	41 (9.3)	39 (8.3)	37 (8.1)
**Heart attack**				
No	1521 (90.9)	520 (93.4)	499 (89.4)	502 (89.8)
Yes	135 (9.1)	37 (6.6)	59 (10.6)	57 (10.2)
**Continuous variables, mean (SD)**				
Waist circumference	99.4 (15.0)	113 (12.9)	98.4 (9.3)	87.1 (9.4)
Height	167 (9.8)	167 (9.6)	167 (10.2)	167 (9.7)
Weight	79.5 (19.2)	97.0 (17.3)	76.8 (12.0)	64.8 (11.4)

ASMI: Appendicular skeletal muscle mass index.

**Table 2 cancers-15-01081-t002:** Association between DXA measures of body composition and risk of all-cause and cancer-specific mortality in cancer survivors.

	No. Death/Person-Years	Age-Adjusted HR (95% CI)	aHR (95% CI)	aHR (95% CI)
		Model 1	Model 2	Model 3
** *All-cause mortality* **				
**Total fat mass**				
Low	169/1602.3	Ref	Ref	Ref
Medium	237/2144.8	1.48 (1.21–1.80)	1.09 (0.89–1.34)	1.09 (0.89–1.34)
High	262/2207.4	1.76 (1.45–2.13)	1.30 (1.06–1.61)	1.31 (1.06–1.61)
**Appendicular skeletal muscle mass**				
Without sarcopenia	253/8624.7	Ref	Ref	Ref
With sarcopenia	155/2422.8	1.58 (1.28–1.94)	1.51 (1.22–1.88)	1.60 (1.28–1.99)
** *Waist circumference* **				
Normal	211/6320.0	Ref	Ref	Ref
High	457/8801.2	0.85 (1.17–0.73)	1.23 (0.98–1.54)	1.01 (0.82–1.47)
**Body Mass Index**				
Normal	219/4450.1	Ref	Ref	Ref
obese	171/5007.4	0.68 (0.58–0.80)	0.71 (0.50–1.10)	0.66 (0.41–1.14)
** *Cancer-specific mortality* **				
**Total fat mass**				
Low	58/4768.8	Ref	Ref	Ref
Medium	72/4416.3	1.04 (0.73–1.47)	1.01 (0.71–1.44)	0.99 (0.70–1.42)
High	83/4195.8	1.42 (1.01–2.00)	1.24 (0.87–1.77)	1.23 (0.86–1.75)
**Appendicular skeletal muscle mass**				
Without sarcopenia	83/7840.7	Ref	Ref	Ref
With sarcopenia	56/1976.5	1.88 (1.33–2.65)	1.74 (1.23–2.29)	1.84 (1.28–2.65)
** *Waist circumference* **				
Normal	66/6320.0	Ref	Ref	Ref
High	147/8801.2	0.72 (0.54–0.96)	0.67 (0.50–0.90)	0.62 (0.43–0.89)
**Body Mass Index**				
Normal	82/4737.7	Ref	Ref	Ref
Overweight or obese	126/10,104.8	0.63 (0.48–0.83)	0.58 (0.43–0.78)	0.51 (0.35–0.73)

Abbreviations: aHR: adjusted hazard ratio, CI: confidence interval. Mean fat mass; low = 18,571.7 g, moderate = 27,678.4 g, and high = 41,946.0 g. Model 1: The model adjusted for age. Model 2: The model adjusted for the following covariates: age, sex, race, marital status, education, energy intake, smoking status, comorbidity burden, and history of more than 1 cancer (for cancer survivors). Model 3: The model adjusted for the same covariates as Model 2 and additionally corrected for NHANES sampling weight.

**Table 3 cancers-15-01081-t003:** Association between DXA measures of body composition and risk of all-cause and cancer-specific mortality in obesity and non-obesity-related cancer survivors.

Measures of Body Composition	Obesity-Related Cancers (n = 699)		Non-Obesity-Related Cancers (n= 983)	
	Model 1 Age Adjusted HR (95 CI)	Model 2 Multivariable HR (95 CI)	Model 1 Age Adjusted HR (95 CI)	Model 2 Multivariable HR (95 CI)
** *All-cause mortality* **				
**Total fat mass**				
Low	Ref	Ref	Ref	Ref
Medium	1.14 (0.84–1.54)	1.26 (0.93–1.71)	1.02 (0.77–1.35)	0.99 (0.74–1.32)
High	1.81 (1.34–2.43)	1.69 (1.23–2.32)	1.27 (0.97–1.67)	1.14 (0.86–1.52)
**Appendicular skeletal muscle mass**				
Without sarcopenia	Ref	Ref	Ref	Ref
With sarcopenia	1.49 (1.09–2.03)	1.53 (1.10–2.13)	1.79 (1.35–2.36)	1.61 (1.18–2.19)
** *Cancer-specific mortality* **				
**Whole body fat mass**				
Low	Ref	Ref	Ref	Ref
Medium	1.09 (0.65–1.81)	1.08 (0.65–1.81)	1.00 (0.60–1.67)	0.99 (0.59–1.67)
High	1.51 (0.91–2.52)	1.42 (0.84–2.41)	1.54 (0.95–2.48)	1.35 (0.81–2.23)
**Appendicular skeletal muscle mass**				
Without sarcopenia				
With sarcopenia	Ref	Ref	Ref	Ref
	1.65 (0.98–2.77)	1.76 (1.03–3.02)	2.23 (1.38–3.62)	1.80 (1.04–3.12)

Abbreviations: aHR: adjusted hazard ratio, CI: confidence interval. Model 1: The model adjusted for age. Model 2: Age, sex, race, education, marital status, smoking status, energy intake, burden of comorbidities, history of more than 1 cancer (for cancer survivors) were adjusted for in this model. For cancer survivors, time elapsed since cancer diagnosis was stratified in this model.

**Table 4 cancers-15-01081-t004:** Associations between DXA measures between total fat mass and sarcopenia with risk of all-cause mortality in cancer survivors by race.

Measures of Body Composition	Whites		Blacks	
	Model 1 Age Adjusted HR (95 CI)	Model 2 Multivariable HR (95 CI)	Model 1 Age Adjusted HR (95 CI)	Model 2 Multivariable HR (95 CI)
** *All-cause mortality* **				
**Total fat mass**				
Low	Ref	Ref	Ref	Ref
Medium	1.27 (0.97–1.56)	1.24 (0.97–1.57)	1.02 (0.98–0.59)	0.78 (0.44–1.38)
High	1.50 (1.18–1.50)	1.37 (1.07–1.75)	1.80 (1.12–1.78)	1.27 (0.73–2.22)
**Appendicular skeletal muscle mass index**				
Without sarcopenia	Ref	Ref	Ref	Ref
With sarcopenia	1.57 (1.23–1.98)	1.53 (1.19–1.95)	2.76 (1.44–5.32)	2.99 (1.39–6.06)

Abbreviations: aHR: adjusted hazard ratio, CI: confidence interval. Mean total fat mass for Black people = 31,234.8 g and White people = 28,958.0. Model 1: The model adjusted for age. Model 2: In addition to age, the following covariates were adjusted for in this model: sex, race, education, marital status, smoking status, energy intake, burden of comorbidities, history of more than one cancer (for cancer survivors). This model was stratified by time elapsed since cancer diagnosis for cancer survivors.

## Data Availability

The datasets generated and/or analyzed during the current study are publicly available from the Centers for Disease Control and Prevention (https://wwwn.cdc.gov/nchs/nhanes/Default.aspx (accessed on 8 June 2022)).

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
