# Peer review of "Associations of Total Body Fat Mass and Skeletal Muscle Index with All-Cause and Cancer-Specific Mortality in Cancer Survivors"

_cancers, 2023, doi:10.3390/cancers15041081_

Round 1
Reviewer 1 Report
This paper mainly studies the associations of total body fat mass and skeletal muscle index with all-cause and cancer-specific mortality in cancer survivors, and analyzes the effects of fat、sarcopenia、and race on the risk of death, which have significant clinical and public health implications, however, there are some issues with this article. My suggestion is as follows:
Main points
1. The title of this review is the relationship between skeletal muscle index in cancer survivors, while the article mainly describes the skeletal muscles of the limbs. There is a discrepancy between the two, and it is recommended to revise;
2. The abstract part includes methods, results, and conclusions. It is recommended to advance the Abstract by one line, and change the first part of the abstract to the purpose;
3. This article describes the skeletal muscle index many times, but there is no keyword part, it is recommended to add;
4. In the introduction part, the author mainly introduces the impact of obesity on cancer, and does not introduce the changes of skeletal muscle on cancer. In addition, the author should also introduce what is all-cause mortality and cancer-specific mortality. It is recommended to modify ;
5. In terms of result discussion, it is recommended that the author present each type of data in points and sections;
6. In Table 2, aHR appears for the first time, and the full name is not marked, and the typesetting of Table 2 is a bit confusing. It is recommended to modify it;
7. It is suggested that in the discussion part, add some shortcomings of this study due to the small sample size and short follow-up time;
8. Because the abstract mainly mentions three aspects of total fat mass, sarcopenia, and race, it is recommended to focus on these three points in terms of structure.
Secondary points:
1. There are too many keywords, generally 3-5 are recommended;
2. The titles of Table 2 and Table 3 are exactly the same, but the contents of the tables are completely different. It is recommended to modify the title;
3. The abscissa and ordinate variables in Figure 2 have no units;
4. There are some minor grammatical errors in the text, it is recommended to modify;
5. There is an error in the typesetting of the references. If the names are separated by commas instead of semicolons, it is recommended to modify them.
Reviewer 2 Report
In their Paper, "Associations of total body fat mass and skeletal muscle index with all-cause and cancer-specific mortality in cancer survivors," Aduse-Poku et al. provide a clear presentation of anthropometric data from cancer patients obtained and analyzed from the NHANES databases ('99-'06 and '11-'18). The results generally support the idea that both adiposity and sarcopenia are risk factors for mortality among different classifications of cancer and possibly affect different racial groups. Their use of DXA scan data, together with disease outcomes ensures that this will be considered one of the stronger, more reliable studies investigating these relationships. However, there are a few recommendations that I believe will improve the manuscript:
1. MAJOR: The primary table presented by the authors (Table 1) presents the number of cancer survivors among different group classifications. While the number of cancer survivors is necessary to present for different groups, as it provides the hazard ratios, there is very little descriptive data of the groups. For example, despite clearly stating that groups were divided into terciles for fat mass (LOW vs MOD vs HIGH), there is no indication of what the cut-off values for the different groups ended up being. Along these lines, it would be useful for the readers to have the cut-offs, as well as the mean data, to help make sense of some of the variability in hazard ratios. For example, by including mean total fat masses for the three groups when looking at the association of sarcopenia and mortality in blacks and whites, the readers may be able to try to make sense of possible differences and propose causes/mechanisms to explain such differences.
2. MAJOR: Along the lines of #1 (above), the authors should explore their differences in the context of mechanisms. I understand that as an observational, epidemiological study, this is not the proper model to conclude what the cause-and-effect relationships are, but the authors do not even provide possible reasons for differences between groups. The conclusion reads like an epidemiological review article, rather than a dissemination of their own data. By including the actual data values (either in primary text or supplemental), the authors could have more to talk about and help direct future research that might look at taking it a step further to explore cause-and-effect relationships. For example, WHY do the authors suppose that the risk of mortality is so much higher in black sarcopenic patients compared to their white counterparts?
3. MINOR: On line 110, the authors describe their Appendicular Skeletal Muscle Index (ASMI) as skeletal muscle mass/height. I understand that the authors are trying to highlight that ASMI includes a height adjustment to skeletal mass measures, but the way it is presented is misleading. It should better be shown that ASMI is an index of skeletal muscle mass adjusted to height2 and not just height (which is how it reads). As alluded to in both #1 and #2 above, seeing the actual data of these kinds of measures will also help clarify to the reader what range of data actually comprised the terciles for fat mass and sarcopenia grouping for different sub-groups.
